# On-Site Determination of Methylmercury by Coupling Solid-Phase Extraction and Voltammetry

**DOI:** 10.3390/molecules27103178

**Published:** 2022-05-16

**Authors:** Paolo Inaudi, Elio Mondino, Ornella Abollino, Mery Malandrino, Monica Argenziano, Laura Favilli, Roberto Boschini, Agnese Giacomino

**Affiliations:** 1Department of Drug Science and Technology, University of Torino, Via Pietro Giuria 9, 10125 Turin, Italy; ornella.abollino@unito.it (O.A.); monica.argenziano@unito.it (M.A.); laura.favilli@unito.it (L.F.); 2Department of Chemistry, University of Torino, Via Pietro Giuria 7, 10125 Torino, Italy; elio.mondino@edu.unito.it (E.M.); mery.malandrino@unito.it (M.M.); 3FKV SrL, Largo delle Industrie, 10, 24020 Torre Boldone, Italy; r.boschini@milestonebg.onmicrosoft.com

**Keywords:** mercury, methylmercury, fish product, on-site analysis, stripping voltammetry

## Abstract

A measurement and speciation procedure for the determination of total mercury (Hg_TOT_), inorganic mercury (Hg_IN_), and methylmercury (CH_3_Hg) was developed and the applicability for on-site determination was demonstrated. A simple, portable sample pretreatment procedure was optimized to extract the analytes. Home-made columns, packed with a new sorbent material called CYXAD (CYPHOS 101 modified Amberlite XAD), were used to separate the two forms of the analyte. Hg_TOT_ and CH_3_Hg were determined by anodic stripping voltammetry (ASV), using a solid gold electrode (SGE). Two certified reference materials (BCR-463 Tuna Fish and Tuna Fish ERM-CE 464) and eight fresh fishes were analyzed. Then, the results that were obtained following the optimized portable procedure were compared with the concentrations obtained, using a direct mercury analyzer (DMA). This quantification, using the two techniques, demonstrated the good performance of the proposed method.

## 1. Introduction

In aquatic systems, bacteria transform inorganic mercury, Hg_IN_, into methylmercury, CH_3_Hg [1]. CH_3_Hg is considered the most toxic form of Hg and is very harmful due to its ability to bioaccumulate. As a result of its biomagnification capacity, organisms at the top of a food chain are especially exposed [2], but CH_3_Hg is present in varying amounts in all seafood. In fact, for humans, fish is the main source of mercury and methylmercury in the diet. CH_3_Hg binds to the sulfhydryl groups of amino acids and is absorbed from the gastrointestinal tract. Furthermore, it is able to cross the blood-brain barrier [3]. For this reason, due to their developing brain, infants and children are considered the most at risk of CH_3_Hg toxicity. Health problems such as numbness around the mouth or limbs, difficulty thinking clearly, hair loss, stomach pain and fatigue have been reported in cases involving the consumption of fish with high mercury content. People who frequently eat fish, as well as fauna that feed mainly on fish, are highly exposed to the risks associated with the bioaccumulation of CH_3_Hg. For many populations, fish make a vital contribution to the diet in terms of survival and health, particularly in many developing countries. For some populations, for example, in Asia, it is estimated that 75% of the daily proteins ingested as part of the diet are derived from the consumption of fish [4]. They provide essential nourishment, especially quality proteins and fats, vitamins and minerals. Furthermore, fish is a source of income from the fish trade, which can, consequently, enable the purchase of other food products [5,6]. In European Union legislation, the maximum levels of total mercury (Hg_TOT_) are 0.5 mg kg^−1^ for small and medium fishes, mussels and most forms of seafood, and 1 mg kg^−1^ for predatory fish (e.g., tuna and swordfish) [7], while a maximum limit of CH_3_Hg in food has not as yet been established. A provisional tolerable weekly intake (PTWI) of 1.3 µg kg^−1^ CH_3_Hg and 4 µg kg^−1^ Hg body weight has been set [8]. For all these reasons, it is essential to monitor the concentration of Hg and its distribution, in its inorganic and organic forms, in fish products [9]. Hg_TOT_ can be quantified by cold-vapor atomic fluorescence spectrometry (CV AFS) [10], cold-vapor capacitively coupled plasma micro-torch optical emission spectrometry (CV-µCCP-OES) [11], cold-vapor atomic absorption, or high-resolution inductively coupled plasma mass spectrometry [12] and capillary electrophoresis (CE) [13]. All these techniques involve long sample pretreatment methods, high management costs and specialized personnel. Gas chromatography (GC) with various detectors, relying on mass spectroscopy (MS) [14], electron capture detection [15], atomic fluorescence spectrometry [16] and high-performance liquid chromatography (HPLC), coupled with atomic emission detection [17] or coupled with high-performance liquid chromatography [18], have been used to determine CH_3_Hg. For example, isotope dilution-GC/ICP-MS [19] and purge-and-trap GC-MS [20] have also been reported as analytical tools. In the literature, some papers also report the application of electrochemical techniques for the determination of CH_3_Hg [21,22], such as potentiometric stripping analysis (PSA), current stripping chronopotentiometry (CSP) [23] and, in particular, voltammetry [24]. Anodic stripping voltammetry (ASV) was used for the determination of CH_3_Hg at concentrations starting from 2 × 10^−8^ mol L^−1^ [25]. To our knowledge, the first example of the application of ASV to the determination of CH_3_Hg was based on a linear calibration diagram obtained using a mercury drop electrode, which allowed the researchers to obtain a linear response in a concentration range from 10^−4^ to 10^−7^ mol L^−1^ [26]. The method of double standard additions was used for the quantification of CH_3_Hg. Agraz et al. reported the use of a particular modified carbon paste electrode for the determination of Hg^2+^ and CH_3_Hg after a pre-concentration period, with a detection limit (DL) of 1 × 10^−8^ mol L^−1^ [27]. Differential pulse voltammetry (DPV), coupled with a glassy carbon electrode modified with Nafion, was tested for CH_3_Hg analysis, with a DL equal to 4.5 × 10^−8^ mol L^−1^ [28]. The DL dropped to 4.5 × 10^−11^ mol L^−1^ using a multiple square-wave voltammetric technique; the lowest determined concentration of CH_3_Hg in this study was 4 × 10^−8^ mol L^−1^. The reduction of CH_3_Hg and its subsequent re-oxidation is the most exploited voltammetric technique for the quantification of this analyte. Most notably, a detection limit of 5.6 × 10^−7^ mol L^−1^ was described using conditions optimized for the determination of CH_3_Hg in dogfish muscle samples [29]. In another study [22], carbon microelectrodes were used for the analysis of CH_3_Hg in chloride media. A positive response to the technique was also demonstrated at nanomolar concentrations for CH_3_Hg, but the linearity of the method was not optimal [30]. Several studies using mercury [29] or gold [31] film electrodes were carried out. The use of polymer-coated carbon electrodes [28] or gold nanoparticle-modified electrodes [32] allowed researchers to lower the DL to the nanomolar concentration range. Korolczuk and Rutyna proposed a method for analyzing the organic forms of mercury by eliminating the interference of inorganic forms, complexing Hg^2+^ ions with diethylenetriaminepentaacetic acid (DTPA): in this way, a selective preconcentration on the electrode is obtained by applying a reduction potential to the metallic state to more negative values than the potential of the CH_3_Hg^+^ reduction to elemental mercury [31].

The purpose of this paper is to highlight the applicability of a patented procedure with freshly caught fish for the determination of Hg_TOT_, Hg_IN_ and CH_3_Hg content via ASV outside laboratories. The work was mainly focused on facilitating the portability and ease of execution of the different steps involved in the treatment of the solid samples, along with the voltammetric determination of the analytes in the obtained sample solution, using a simple solid gold electrode. Two certified reference materials (BCR-463 Tuna Fish and Tuna Fish ERM-CE 464) and eight fresh fish samples were tested. A sample analysis was also conducted using a direct mercury analyzer (DMA), considered a reference method for Hg_TOT_ and CH_3_Hg determination [33], in order to compare the results of the two methods.

## 2. Results

In our previous work, we optimized an easy procedure for mercury speciation in the laboratory. In the present study, the procedure was further improved and its applicability for outside analysis was assessed.

The procedure, once further improved and applied on-site, permitted us to obtain performance data in terms of repeatability, linearity, accuracy and detection limits comparable to those previously obtained in the laboratory. A deposition time of 120 s was found to be the best option for analyte concentrations of up to 50 µg L^−1^. In a blank matrix, the LOD of Hg_IN_ and CH_3_Hg was expected to show a 3σB/slope; it was 0.40 µg L^−1^ for Hg_IN_ and 0.50 µg L^−1^ for CH_3_Hg, while sensitivity was 1.71 µA/µgL^−1^ and 1.68 µA/µgL^−1^ for Hg_IN_ CH_3_Hg, respectively. The relative error for the quantification of 1 µg L^−1^ of the analytes was −1% and +1% for Hg_IN_ and CH_3_Hg, respectively.

All the samples were analyzed outdoors, following the entire portable procedure (extraction of the analytes and voltammetric determination) and adopting the proposed kit.

Figure 1a,b shows the voltammograms obtained during the analysis of ERM-CE 464 for the quantification of Hg and CH_3_Hg, respectively.

For Hg determination, the linear regression equation obtained from the standard additions was y = 0.0756x + 0.0018 (R^2^ = 0.9948), while for CH_3_Hg, the equation was y = 0.0651x + 0.019 (R^2^ = 0.9958). The method shows the good linearity of the technique for both considered analytes. The concentrations of Hg_TOT_ and CH_3_Hg found in the reference materials and in the examined fishes are described in Table 1, which shows the results obtained by the portable voltammetric kit and by using DMA.

In the case of Hg_TOT_, recoveries of 96% and 93% were obtained for ERM-CE464 and BCR 463, respectively, in the case of ASV measurement and 96% and 72%, respectively, using DMA. For CH_3_Hg, recoveries of 95% and of 68% were obtained for ERM-CE464 and BCR 463, respectively, in the case of ASV measurement, and 82% and 77%, respectively, using DMA. Then, the portable procedure was applied to the fish samples. As an example, Figure 2a,b shows the voltammograms obtained when analyzing commercial fresh tuna fish for Hg and CH_3_Hg determination, respectively. For all the samples, R^2^ values higher than 0.99 were obtained.

The concentrations of Hg_TOT_ and CH_3_Hg achieved using the proposed portable procedure and DMA are compared in Table 2.

No measurable Hg peaks were detected in the trout samples or in the codfish, while in all the other samples, the concentrations for fresh fish exceeded the legal limits for the predator fish. In fact, all the concentrations were higher than 1 mg kg^−1^, the limit for the predator fish; in particular, both the slices of swordfish that were tested showed an Hg content two to three times higher than the maximum permitted value. These results were confirmed by the results obtained using DMA. The LOQ in the fish matrix, calculated as the minimum quantity determined with good accuracy, was 0.2 mg kg^−1^ with the solid gold electrode (SGE). Our new method can be considered appropriate for testing the mercury content in this matrix since the highest permissible concentration in tuna fish is set to 0.5 mg kg^−1^ _wet fish_ by the European Commission [7]. For the sake of completeness, it should be said that the Hg_IN_ retained in the column was recovered with 5 mL of HNO_3_, and then diluted and analyzed. The recovery was quantitative (comparing the obtained concentrations with those of Hg_TOT_ and CH_3_Hg found in each sample). The results were always compared with the DMA. The data are not reported here since the aim of this work was to optimize the procedure in order to make possible the direct quantification of CH_3_Hg. To better compare the results, Figure 3a,b shows the correlation between the concentrations obtained (analyzing swordfish, tuna fish and blue marlin samples) using the two techniques for Hg and CH_3_Hg, respectively.

A value of R^2^ > 0.99 was obtained for both the analytes, demonstrating the good applicability of the proposed portable procedure for mercury speciation studies.

### Novelty of the Study and Concerns about the Portable Kit

The novelty of this study is the further optimization of the procedure and its on-site application for mercury speciation in fish samples. To our knowledge, this is the first method that permits the treatment of solid samples for the extraction of Hg and CH_3_Hg, separating them before analysis and the determination of their concentration, directly in the field.

The most significant improvements have been: (i) the use of piston syringes, which permits carrying out the procedure manually, without the aid of supports or pumps. Moreover, this type of syringe can be packed with CYXAD and then stored at room temperature for up six months. (ii) The possibility of determining CH_3_Hg in the sample solutions after elution. In this way, the elution of Hg_IN_ that was retained in the CYXAD cartridge was not required, reducing the number of solutions that need to be carried in the field and speeding up the procedure.

The modifications simplified the applicability of the procedure for on-site analysis while maintaining the same performance, in terms of accuracy and precision, as that obtained in the laboratory.

The transportable battery assures 15 h of autonomy, with the battery recharging overnight. Consequently, the possibility of performing analyses on subsequent days is secured.

The use of a mini dry bath to warm the samples during the extraction step permits more uniform heating; the dissolution of the sample appears more homogeneous, consequently facilitating the filtration step; furthermore, the heating takes place in dry conditions, reducing the volume of high-purity water (HPW) needing to be brought into the field. In order to reduce waste, it is important to use small volumes of the reagents for both the analysis and pre-treatment of the sample. In particular, additions of mercury standards should be made in small volumes to the voltammetric cell, in order to reduce the volumes of solutions containing mercury; for this reason, it is essential that they are collected and transported to the laboratory for proper disposal. In our previous study, the direct determination of CH_3_Hg did not seem possible since the concentration of chlorides and residues of organic matter in the solution eluted by the CYXAD were too high: chlorides could cause damage to the gold surface of the electrode, while organic matter created problems during the phase of the deposition of the analyte onto the electrode. Therefore, CH_3_Hg concentration was determined by the difference between Hg_TOT_ and Hg_IN_. Since Hg_IN_ was previously retained by the column and had to be eluted with concentrated HNO_3_, there was higher variability in the recovery of the analyte and in its quantification. In this work, after elution, the solution was treated with H_2_O_2_ to oxidize most of the organic matter. Moreover, the analysis was performed using the “medium exchange” technique. These procedures reduce the passivation phenomena on the electrode’s active surface due to the deposition of other substances on the WE surface and increase the sensitivity of the method.

## 3. Materials and Methods

### 3.1. Instruments and Reagents

A mini dry bath, coupled with a metal block for 2 × 50 mL tubes, was used in the pre-treatment of the samples (Starlab S.r.l, Milano, Italy). A PalmSens4 portable potentiostat (PalmSens, Houten, the Netherlands) was adopted for the electrochemical analyses. It was then connected to a portable computer; the different parameters for the analysis were set using the PSTrace 5.8 software. A magnetic stirrer (IKA-Topolino, Staufen, Germany) was connected to the PalmSens4 and was powered by a portable battery. An electrochemical cell with 15 mL minimum usable volume (Kit BASI, West Lafayette, IN, USA), equipped with an SGE (peek OD: 6 mm, ID: 1.6 mm, glass-bodied; Ionode, Tennyson, Australia) as a working electrode, a Pt auxiliary electrode with 6 mm × 6 mm foil (Ionode, Tennyson, Australia) and a glass-bodied refillable reference electrode, single-junction, with Ag/AgCl (Ionode, Tennyson, Australia) was used. A DMA-80 Direct Analyzer (FKV SrL, Torre Boldone, Italy) in the FKV laboratory in Torre Boldone (BG, Italy) was used for the atomic absorption spectrometry analyses. Analytical grade reagents were used. Nitric acid (HNO_3_, 65%, Sigma-Aldrich Merck, Darmstadt, Germany), hydrogen peroxide (H_2_O_2_, 30%, Sigma-Aldrich Merck, Darmstadt, Germany) and hydrochloric acid (HCl, ACS Reagent, 37%, Fisher Scientific Italy, Rodano (MI), Italy) were also used. Briefly, 1000 mg L^−1^ of Sigma-Aldrich Hg solution was used to prepare working standard solutions of 1 mg L^−1^. CH_3_Hg standard solutions were prepared in high-purity water (HPW-Milli-Q, Millipore, 18.2 MΩ cm), using due precautions, from CH_3_HgCl crystals (Pestanal, analytical standard, Sigma-Aldrich Merck, Darmstadt, Germany), then stored in a cool, dry place away from oxidizers. Calibration standards for DMA-80 were prepared using a NIST traceable stock solution of 1000 mg L^−1^ Hg, preserved in 5% HNO_3_. All the working standards were freshly prepared weekly. An ionic liquid, namely, trihexyl(tetradecyl)phosphonium chloride (CYPHOS 101 solution, Sigma-Aldrich) was used to modify a polymeric sorbent, i.e., Amberlite XAD-1180 (Sigma-Aldrich). This new sorbent material (CYXAD: CYPHOS-modified XAD) was used to separate the Hg_IN_ and CH_3_Hg after extraction with HCl.

The patented equipment for the outdoor analysis adopted in this work can be easily transported in a backpack and consists of: (i) test tubes filled with H_2_O_2_ at 30%, HNO_3_ at 65%, or HCl at 37%; (ii) a bottle containing the solution for the electrochemical cleaning (0.1 mol L^−1^ HClO_4_, 1.5 mmol L^−1^ NaCl and 0.5 mmol L^−1^ -Na_2_-EDTA); (iii) two bottles containing HPW; (iv) test tube with 1 mg L^−1^ Hg standard solution; (v) a bottle containing 60 mmol L^−1^ HCl as the supporting electrolyte; (vi) a small balance to weigh aliquots of the sample; (vii) a mini dry bath for sample pretreatment; (viii) a portable battery; (ix) syringes to transfer the aliquot of a solution and filters to filter the aliquot of the sample solutions; (x) cartridges containing the CYXAD phase, equipped with a disposable inlet filter; (xi) micropipettes and tips for transferring the sample solution and adding known concentrations of analytes for calibration by standard additions; (xii) a cell, its perforated cap and the three electrodes (SGE; Ag/AgCl/KCl; Pt); (xiii) a Palmsens4 portable potentiostat; (xiv) a portable computer; (xv) a tank for the wastewater.

### 3.2. Samples

The reference materials (RMs), namely, BCR-463 Tuna Fish ([Hg_TOT_] = 2.85 ± 0.16 mg kg^−1^; [CH_3_Hg] = 3.04 ± 0.16 mg kg^−1^) and ERM-CE464 Tuna Fish ([Hg_TOT_] = 5.24 ± 0.10 mg kg^−1^; [CH_3_Hg] = 5.50 ± 0.17 mg kg^−1^) were analyzed to assess the quality and accuracy of the analytical technique in question. Eight samples, specifically, two slices of tuna fish and one slice each of swordfish, blue marlin, codfish, rainbow trout and salmon trout were analyzed with the aim of testing the applicability of the technique for in situ analysis. The analyses were carried out near the experimental breeding tanks of the Department of Agricultural, Forest and Food Sciences, located in Carmagnola (Turin, Italy), where the rainbow trout was also caught from a tank. The other samples were purchased in local supermarkets or fish-markets located in the province of Turin (Italy). In all cases, the samples were analyzed outdoors to test the applicability of the whole procedure in the field.

### 3.3. Procedures

#### 3.3.1. Extraction of Hg_TOT_

Aliquots of 0.5 g–1 g of RM or the sample were added with a 1:1 mixture of HNO_3_/H_2_O_2_ into 50 mL test tubes and warmed for 20 min at 70 ± 5 °C with the mini dry bath. Afterward, the solution was filtered through 0.45 µm polytetrafluoroethylene (PTFE) syringe filters and diluted to 15 mL with HPW. Subsequently, all the solutions were analyzed. All the analyses were performed in duplicate.

#### 3.3.2. Preparation of the Modified New Sorbent—CYXAD

HPW, HCl (10% *v/v*) and ethanol were used in order to wash the Amberlite XAD-1180 resin to remove inorganic contaminants and residues; afterward, it was left to dry at room temperature. Lastly, it was functionalized using CYPHOS 101 with an resin:IL = 2:1 ratio in 5 mL of ethanol for 6 h. The suspension was then filtered through a PTFE filter and the new sorbent material that was obtained (CYXAD) was dried in an oven (1 h—60 °C), then collected in a vessel.

At this point, 40 mg of CYXAD was inserted into a Combitips 500 µL advanced Eppendorf syringe, as shown in Figure 4 (diameter = 4.5 mm).

Previously, a Teflon filter was inserted into the syringe to maintain the resin inside the tube. The CYXAD material is durable and can be stored for up to six months; this way, it is possible to prepare packed cartridges that are ready for use.

#### 3.3.3. Speciation System with Voltammetry

Each sample was subjected to two different types of pretreatment. The first aliquot was treated as explained earlier for the quantification of Hg_TOT_, while the second aliquot was treated for the speciation study, as follows: 1 g of sample was placed into a 50 mL test tube, in contact with 6 mL of 8 mol L^−1^ HCl, and heated in the mini dry bath (20 min—70 °C). The high concentration of Cl^−^ in the solution permits the formation of a negative complex with Hg^2+^ (tetrachloromercurate (II), [HgCl_4_]^2−^) and a neutral complex with CH_3_Hg (methylmercury chloride, CH_3_HgCl) [34]. Later, to lower the concentration of chloride ions (4 mol L^−1^), the solution was diluted to 1:2 with HPW, at which ratio the highest resin efficiency is realized. The solution was forced through the CYXAD cartridge that retained Hg_IN_ quantitatively, while the CH_3_Hg was eluted. In this step, the CYXAD retains the anionic species of Hg, while the neutral species is eluted. In our former work [24], the eluate containing CH_3_Hg was useless because it had too high a level of Cl^−^, which would damage the SGE. The Hg_IN_ immobilized on the CYXAD was recovered with 5 mL of 6 mol L^−1^ HNO_3_ and the voltammetric analysis was performed. The content of CH_3_Hg was determined by difference ([CH_3_Hg] = [Hg]_TOT_ − [Hg]_IN_). After the elution in this scenario, the solution was added with concentrated H_2_O_2_ (1:1 ratio); then, 0.5 mL was added to 9.5 mL of 60 mmol L^−1^ HCl and analyzed. To reduce the matrix effect, the analysis was made utilizing the “medium exchange” procedure: the deposition step at 0 V was achieved by placing the electrode into the sample solution, then, with the aid of the “hold” function, the potential was kept constant and the cell was changed for a new one containing 20 mL of supporting electrolyte; then, the stripping step was begun.

#### 3.3.4. ASV Measurement

Every day, the SGE was polished with alumina powder and activated by utilizing a potential of +0.6 V for 30 s in 60 mmol L^−1^ HCl [35]. There was no need to repeat these procedures during the analysis. Many researchers use CV treatment as an activation step for solid electrodes, for example, in the case of SGE, using H_2_SO_4_ as the supporting electrolyte; however, in our experience, an activation step of applying a potential of 0.60 V for 60 s in 60 mmol L^−1^ HCl before mercury determination is sufficient to keep the electrode surface active to enhance the quality and reproducibility of the mercury signal. The CV voltammogram, recorded in H_2_SO_4_, presents the characteristic anodic and cathodic peaks at +1.25 V and at +0.90 V, respectively [36]. Cyclic voltammetry is commonly used to check the quality and the area of the electrode surface [37].

After each quantification, the SGE was stored in a solution of 0.2 mol L^−1^ HClO_4_/3 mmol L^−1^ NaCl/1 mmol L^−1^ NaEDTA (0.8 V—30 s), to eliminate the residues of Hg from its surface. For the development of the procedure, Hg standard solutions were used. Firstly, 10 mL of the aliquot test solution of 60 mmol L^−1^ HCl was transferred into the voltammetric cell to register the blank signal. The values of the voltammetric parameters were: frequency 15 Hz, step potential 0.006 V, and amplitude 0.03 V. The standard additions method was used for the determination of Hg content in the samples. After recording the voltammogram of the sample solution when spiked with a known concentration of analyte, the aliquots of Hg at a known concentration were added and the corresponding signals were recorded.

Aliquots of 0.5–1 mL of sample solutions were placed into the cell for the determination of analytes and diluted to 10 mL with the supporting electrolyte. The concentrations of Hg_TOT_ and CH_3_Hg were determined in the sample extract. The content of CH_3_Hg was determined directly in the extractant solution after the addition of H_2_O_2_.

#### 3.3.5. DMA Analysis

All the data acquired using the proposed procedure were evaluated by comparing them with those obtained using DMA [33], the official method for the determination of Hg_TOT_, according to the protocol proposed by J. Calderón [38] for the speciation study. For the quantification of Hg_TOT_, aliquots of each sample were directly analyzed. For the determination of CH_3_Hg, a double liquid-liquid extraction, initially with an organic solvent and then with L-cysteine, was required.

## 4. Conclusions

In this work, the suitability of a patented portable kit for on-site Hg_TOT_ and CH_3_Hg determination in fish samples was demonstrated. SGE is the best option for in situ studies because it does not require surface modification. Since the LOQ (0.2 mg kg^−1^) of the technique is less than the maximum allowable concentration in fish products (0.5 mg kg^−1^), it is possible to use SGE to monitor the concentration of mercury in these matrices. ASV with SGE has proved to be an interesting technique, also in terms of on-site usage, particularly because of its ease of use, portability and sensitivity. The modern portable potentiostat permits on-site analysis. For the determination of Hg_TOT_, an easy treatment with 1:1 HNO_3_:H_2_O_2_ solution using a dry mini-bath allows the user to quantitatively extract mercury from the sample in the field. The major criticality was linked to the speciation step, particularly the need to create a simple, portable and fast technique for the determination of methylmercury. The development of the patented, homemade new resins, CYXAD, permitted us to separate the different forms of mercury: after the extraction of the two forms with HCl, the sample solution was eluted through a cartridge packed with CYXAD. The CH_3_Hg was eluted, while the Hg_IN_ was retained in the cartridge. The direct determination of CH_3_Hg was made possible by using two expedients that facilitate the determination of content during one step of the voltammetric analysis: the addition of H_2_O_2_ to degrade the organic matrix and the use of the medium exchange technique to avoid the electrode passivation effect due to the elevated concentration of chlorides in the eluent solution. With this method, the certified samples were first analyzed with the aim of assessing the accuracy of the technique; subsequently, the applicability of the portable procedure was evaluated by analyzing genuine samples such as fresh tuna steaks, swordfish and trout. The portable procedure has shown excellent applicability for the on-site determination of Hg_TOT_, CH_3_Hg and/or Hg_IN_ since the results obtained with it are consistent with those obtained with DMA.

The new CYXAD cartridges permit the simplification of the application of the procedure out of doors. This cartridge could also be used in the laboratory for the separation of mercury and methylmercury before analysis. For example, the researcher could substitute the long procedure for methylmercury extraction, avoiding the use of organic solvent, e.g., before DMA analysis, making the separation procedures both more eco-friendly and faster.

With regard to the mercury content found in the considered samples, the situation appears worrying, in particular with regard to the sea fish, which showed concentrations above the legal limit for predatory fish. This underlines the importance of having a fast and simple technique that allows researchers to increase the number of controls on these matrices. The proposed procedure meets these needs precisely: it proved to be suitable for rapid screening analyses, permitting an increase in the monitoring of Hg_TOT_ and CH_3_Hg in fish products to protect the health of consumers.

## 5. Patents

The “Portable kit for mercury speciation analysis” obtained an Italian patent license (priority n° 102019000005904).

## Figures and Tables

**Figure 1 molecules-27-03178-f001:**
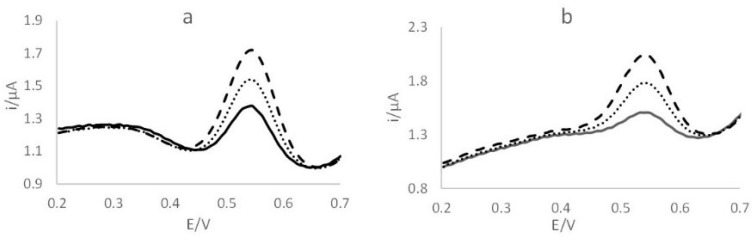
Voltammograms obtained using the portable kit for (**a**) Hg and (**b**) CH_3_Hg quantification: —, ERM-CE 464; ‧‧‧‧ , first addition (5 µg L^−1^); ----, second addition (10 µg L^−1^).

**Figure 2 molecules-27-03178-f002:**
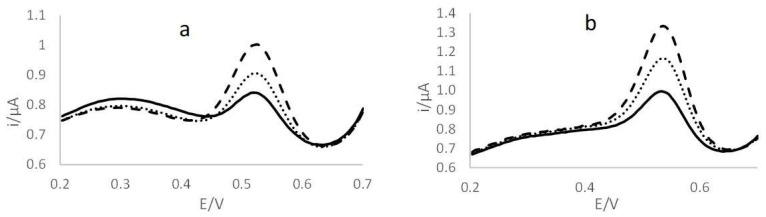
Voltammograms obtained using the portable kit for (**a**) Hg_IN_ and (**b**) CH_3_Hg quantification: — tuna fish; ‧‧‧‧ first addition (5 µg L^−1^); ---- second addition (10 µg L^−1^).

**Figure 3 molecules-27-03178-f003:**
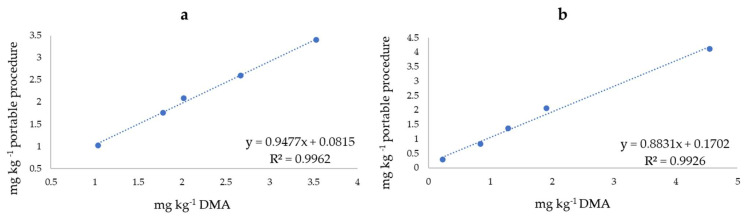
Correlation between the concentrations (mg kg^−1^) of (**a**) Hg_TOT_ and (**b**) CH_3_Hg obtained using the portable procedure and DMA. ● = samples (tuna fish 2, tuna fish 1, blue marlin, swordfish 1 and swordfish 2, reported in ascending order of concentration).

**Figure 4 molecules-27-03178-f004:**
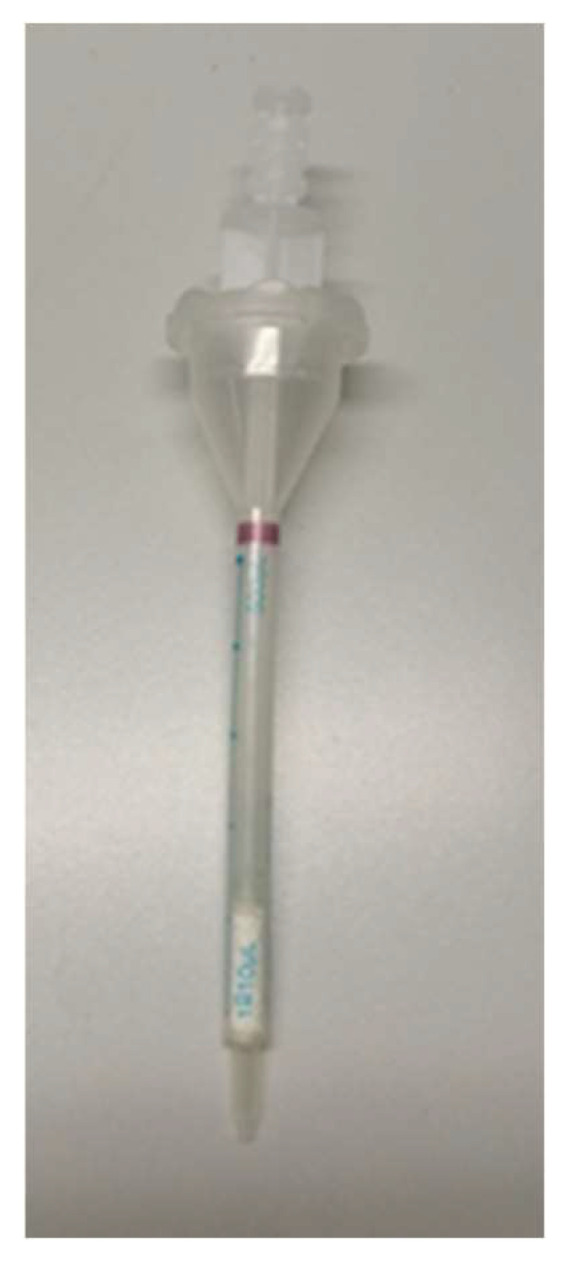
Eppendorf syringe packed with CYPHOS.

**Table 1 molecules-27-03178-t001:** Concentrations (mg kg^−1^) and recoveries (%) obtained for Hg_TOT_ and CH_3_Hg for the analysis of certified materials (tuna fish).

Hg_TOT_					
Sample	[Hg]_TOT certified_	ASV	Recovery	DMA	Recovery
ERM-CE 464	5.24 ± 0.10	5.03 ± 0.04	96	5.04 ± 0.01	96
BCR 463	2.85 ± 0.16	2.65 ± 0.44	93	2.06 ± 0.03	72
**CH_3_Hg**					
**Sample**	**[CH_3_Hg]_certified_**	**ASV**	**Recovery**	**DMA**	**Recovery**
ECM-CE 464	4.89 ± 0.16	4.65 ± 0.04	95	4.01 ± 0.04	82
BCR 463	3.04 ± 0.16	2.08 ± 0.37	68	2.34 ± 0.01	77

**Table 2 molecules-27-03178-t002:** Concentrations (mg kg^−1^) obtained for Hg_TOT_ and CH_3_Hg in fish samples analyzed with voltammetry and by DMA.

Sample	Hg_TOT_		CH_3_Hg	
	Portable Procedure	DMA	Portable Procedure	DMA
Swordfish 1	2.67 ± 0.34	2.60 ± 0.03	4.55 ± 0.04	4.11 ± 0.04
Swordfish 2	3.53 ± 1.52	3.40 ± 0.01	1.91 ± 0.14	2.06 ± 0.01
Tuna fish 1	1.78 ± 0.41	1.76 ± 0.02	1.29 ± 0.23	1.36 ± 0.02
Tuna fish 2	1.04 ± 0.30	1.02 ± 0.02	0.84 ± 0.20	0.82 ± 0.01
Blue marlin	2.02 ± 0.13	2.09 ± 0.05	0.23 ± 0.01	0.29 ± 0.01
Cod fish	<0.20	0.08 ± 0.01	<0.20	0.07 ± 0.01
Rainbow trout	<0.20	0.04 ± 0.01	<0.20	0.02 ± 0.01
Salmon trout	<0.20	0.2 ± 0.01	<0.20	0.08 ± 0.01

## Data Availability

Data, associated metadata, and calculation tools are available from the corresponding author (paolo.inaudi@unito.it).

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
