# Peer review of "On-Site Determination of Methylmercury by Coupling Solid-Phase Extraction and Voltammetry"

_molecules, 2022, doi:10.3390/molecules27103178_

Round 1

Reviewer 1 Report

In this work the authors highlighted the applicability of a patented procedure for the determination of HgTOT, HgIN and CH3Hg content outside laboratories in freshly fished fish by ASV. Two certified reference materials (BCR-463 Tuna fish and Tuna Fish ERM-CE 92 464), and eight fresh fish samples were tested. The contents are interesting, and based on the contents I recommend for publication after some revision.
Special comments:
1. The equation in line 131 needs to be checked and modified.
2. The format of line 186 needs to be checked and modified.
3. The instruments and reagents in line 211 need to simplify the description.
4. It is interesting to discuss the electrode properties during the experimental processes, or the authors are suggested to read and cite the following papers to discuss them in the revised form such as:
Microchemical Journal (2020), 155: 104767, doi: 10.1016/j.microc.2020.104767;
Biosens Bioelectron (2020), 148, 111819, doi: 10.1016/j.bios.2019.111819.

Reviewer 2 Report

The manuscript “On-site determination of methylmercury by coupling solid phase extraction and voltammetry” presents a new method of determining methylmercury in fish and other substrates with high Hg concentrations on-site. There are a few methods out there to determine in-situ high total Hg concentrations in different substrates. However, to my knowledge, this is among the first methods for measuring methylmercury on-site. Still, anodic stripping voltammetry is not a novel procedure, not even for detecting methylmercury. Additionally, the same group published a similar manuscript recently. The differences between the published manuscript and the previous one seem to be simply an improvement of the method previously developed (Giacomino et al. 2021).  Unfortunately, the manuscript does not offer a comparison with such earlier publications making it very hard to accept as an improvement. Their method is sound and was correctly evaluated against reference materials and samples also analyzed by DMA (a well-validated method). However, the absence in the comparison of the previous version of this method is hard to judge how significant is the improvement and if it deserves publication on its own. I suggest the authors make such a comparison before publication or publish this modification to the method as a part of another publication making an application of such modification of the methodology proposed earlier (Giacomino et al. 2021).    

General observations:

  • Writing in the results section needs improvement. Although you have a similar earlier publication, this publication must be able to stand on its own.
  • If this is an improvement of a previously developed and described method (Giacomino et al. 2021), a comparison of both becomes a must that should be included in this manuscript.

Specific observations:

  • In lines 31-40, there seem to be some references missing. There is specific information provided that needs to be supported.
  • In line 98 the writing is confusing. It is Ok to cite previous results, but we need to read your previous publication to understand anything of these sentences.
  • Lines 101-114 seem to belong to methodology.
  • Figure 3. The font in the figure is too small, and the figure itself seems to have low quality.
  • Starting at line 240 the list is a good idea for the manual of your kit, but it could be compressed into a paragraph for the publication.

Round 2

Reviewer 2 Report

Thank you for all the improvements. I still think a case study to demonstrate the method would have been better, but this way is also ok.